# Loco3D: Indoor Multiuser Locomotion 3D Dataset

## Abstract

In the context of human-AI interaction, modeling human actions is a critical and challenging endeavor, with locomotion being a particularly fundamental behavior for AI agents to understand. Modeling human trajectories in complex indoor scenes requires an understanding of how humans interact with their surroundings and other humans. These interactions are influenced by a range of factors, including the geometry and semantics of the scene, the socio-cultural context, and the task each human needs to perform. Previous research has shared datasets containing human motion and scene structure in indoor scenes, but these datasets are limited in scale due to the difficulty and time required to collect data at different locations. To solve the scale problem, we propose to use a virtual reality (VR) system to build a human motion dataset. Specifically, we present Loco3D, a dataset of multi-person interactions in over 100 different indoor VR scenes, including 3D body pose data and highly accurate spatial information. The dataset can be used to build AI agents that operate in indoor environments, such as home robots, or to create virtual avatars for games or animations that mimic human movement and posture. Our evaluation demonstrates that models trained with our dataset have improved multi-person trajectory synthesis performance on real-world data.

## 1 Introduction

The next generation of adaptive systems will involve humans and AI agents working together. To enable this type of collaborative work, AI agents must be embedded with practical models of human behavior that allow them to learn, understand, anticipate, and respond to humans in a truly seamless manner. In particular, locomotion is one of the most fundamental behaviors required of AI agents to perform various actions. However, to design human-like trajectory behaviors in AI agents, we need to synthesize human trajectories in complex environments, such as the home environment. This is challenging because trajectories are affected by the surrounding scene information, including geometry, semantics, dynamics, and social mannerisms, and collecting such data is time-consuming.

There are several prior works that focus on modeling human trajectory through mathematical approaches (Hart et al., 1968; Van Den Berg et al., 2011). While these approaches work well for reaching a goal by avoiding collisions with objects, it is difficult to model human-specific principles, such as social etiquette. As an example, if a person was trying to reach a goal, the most direct path may involve crossing in front of another person who is watching TV. However, doing so would likely obstruct that person's view and be disruptive. Therefore, the person would likely avoid this path, even if it means taking a less direct route.

Along with the development of machine learning techniques, newer works have used data-driven methods. They have successfully synthesized or predicted human trajectory considering scene information using outdoor aerial image datasets (Ess et al., 2007; Zhou et al., 2012; Robicquet et al., 2016) or spacious public buildings (Rudenko et al., 2020; Vendrow et al., 2023). However, this type of imagery is difficult to apply to indoor spaces due to the difference in scale and complexity. For small indoor scenes, it is also challenging to build a dataset for learning scene-aware human trajectory because the camera cannot accurately capture the human-scene interaction due to blind spots caused by obstacles, such as furniture or pillars. Meanwhile, several recent works have addressed this issue using state-of-the-art 3D scanning systems to obtain overall scene information, including human motion and scene geometry in indoor scenes (Hassan et al., 2019; Cao et al., 2020; Guzov et al.,

2021; Zheng et al., 2022; Zhang et al., 2022). However, these datasets only have a small number of scenes because it is expensive and time-consuming to collect such data in the real world.

To address this issue, we propose to use a virtual reality (VR) system that can allow us to obtain accurate full spatial information in a large variety of indoor scenes with high efficiency. A VR setup allows us to conduct data collection in various scenes efficiently. It also allows us to capture accurate human-scene interactions in 3D space since all spatial information is held digitally in virtual space.

In this work, we built a VR system to capture task-oriented human locomotion behavior and constructed a dataset of human trajectories in a variety of home environments. To our knowledge, this is the first dataset that focuses on human trajectory in complex indoor environments, containing multi-person interactions with full 3D body motions and highly accurate full spatial information in over 100 indoor scenes. Our dataset can be used for building AI agents that operate in home environments, such as home robots, to learn human-like trajectories that take into account the geometry and semantics of surrounding objects and social mannerisms. Our main contributions are as follows.

(1) A human behavior data collection system using VR to enable the collection of full 3D scene geometry and multi-person full-body tracking data in a much more efficient manner compared to conventional motion data collection in physical space.

(2) The first dataset, Loco3D, focuses on multi-person trajectories in complex indoor environments. The dataset includes over 7000 multi-person trajectories in 130 indoor environments with 3D pose and 3D scene geometry with semantics.

(3) A novel UNet-based framework that considers the complex scene geometry and other people's movements in indoor scenes to synthesize human-like trajectories.

(4) Experiments on a real-world dataset to demonstrate improved performance on trajectory synthesis using our dataset, Loco3D, compared with other indoor datasets.

## 2 RELATED WORK

### 2.1 HUMAN TRAJECTORY DATASETS

Existing human trajectory datasets can be categorized as human trajectories in outdoor and indoor scenes. A significant amount of prior work has focused on building human trajectory datasets in outdoor scenes, such as Grand Station (Zhou et al., 2012), SDD (Robicquet et al., 2016), ETH (Ess et al., 2007), and TrajNet++ (Kothari et al., 2021). These datasets used aerial cameras to capture human trajectories and surrounding scene images. They learn how humans move by considering other humans' motions and the environment's constraints. However, these approaches cannot be directly applied to learning human trajectories in indoor scenes because indoor scenes are much more complex in terms of geometry and semantics, and the scale of indoor environments is much smaller than outdoor environments. For indoor scenes, THOR (Rudenko et al., 2020) and JRDB (Vendrow et al., 2023) collected human trajectories using a Lidar installed on a robot. However, the scenes were set in an open public space, similar to outdoor scenes, with incomplete scene data due to the Lidar blind spots. Additionally, these indoor datasets are limited in scale as acquiring scene variation is challenging and time-consuming. Loco3D aims to solve the scale problem of prior indoor human trajectory datasets using a VR data collection system.

### 2.2 HUMAN MOTION DATASETS

Understanding 3D human motion is important for various research problems, such as synthesizing motion for virtual scenes. Prior work has studied the 3D human motion problem and built datasets to support research in this domain. Human3.6M (Ionescu et al., 2013), 3DPW (Von Marcard et al., 2018), Panoptic (Joo et al., 2015), and AMASS (Mahmood et al., 2019) are the main large datasets containing human motion without scene interactions. Human3.6M is a large-scale motion dataset containing a wide variety of daily actions of single subjects. It uses human motion capture technology and depth data to obtain accurate motions. 3DPW uses a single camera and a set of Inertial Measurement Units (IMUs) to estimate multi-person 3D poses in the wild. Panoptic focuses on capturing a group of people engaged in social interactions and handles occlusions. AMASS unifies 15 different motion capture datasets and comprises more than 11000 human motions of single subjects.

Another line of research has explored human-scene interactions and focused on understanding how environmental constraints affect human behavior. For example, PROX (Hassan et al., 2019) covers human motions in 12 scenes, and GIMO (Zheng et al., 2022) contains human motions in 19 scenes. HPS (Guzov et al., 2021), EgoBody (Zhang et al., 2022), and LAMA (Lee & Joo, 2023) use head-mounted devices or body-attached IMUs to record motion. A challenge faced by human-scene interactions is that scene variation is difficult and expensive. GTA-IM (Cao et al., 2020) worked on addressing this challenge by using a game engine to synthesize human motions in 10 scenes. HUMANISE (Wang et al., 2022) proposed a language-to-motion generation model to generate human motions in more than 600 scenes. CIRCLE (Araújo et al., 2023) collected over 7000 sequences of daily motions interacting with objects in nine scenes through a VR platform.

The limitations of existing human motion datasets are two-fold: (1) These datasets focus mainly on single-person motions; few datasets have been collected for multi-person interactions. However, understanding multi-person interactions is critical for many real-world use cases, such as robot task assistance and human-robot collaboration. (2) Scene variation is difficult and expensive if a dataset is collected from real environments. Although prior work has used synthesized human motions to tackle this challenge, these motions do not necessarily follow human behavior principles, especially when social interactions between humans are involved. To fill these research gaps, Loco3D is a new dataset containing two people's motions and interactions in indoor environments, collected using VR to allow a large number of scene variations.

### 2.3 HUMAN TRAJECTORY SYNTHESIS

Human trajectory synthesis techniques can be classified as model-based and model-free. Model-based methods include rule-based and optimization-based algorithms, and model-free techniques mostly refer to deep learning approaches. ORCA (Van Den Berg et al., 2011) is a model-based method. It proposed an approach to reciprocal collision avoidance by letting each robot take half the responsibility to avoid collision with thousands of other simulated robots. Although the synthesized trajectories were demonstrated to be smooth, the rules used for trajectory synthesis focused on collision avoidance between robots do not consider the social principles in human collision avoidance, such as respecting social distance or not crossing in front of someone watching TV.

In recent years, we have seen growing interest in model-free methods using deep learning (Gupta et al., 2018; Huang et al., 2019; Mohamed et al., 2020; Wang et al., 2021; Xu et al., 2022; Gu et al., 2022). Among them, YNet (Mangalam et al., 2021) is one of the state-of-the-art approaches that uses a UNet (Ronneberger et al., 2015) to tackle the long-term trajectory synthesis problem in outdoor scenes. However, this work only explores single-person trajectory prediction without considering social interactions in a multi-person trajectory synthesis scenario. NSP (Yue et al., 2022) proposed to incorporate a physics model into the trajectory prediction algorithm based on YNet. The physics model produced a strong inductive bias in modeling pedestrian behaviors and significantly improved the trajectory synthesis performance. However, the physics model emphasizes local behaviors but fails to model the global environmental constraints, which is essential for long-term trajectory prediction.

Our model aims to address the limitations of prior human trajectory synthesis models by considering both the social interactions between humans and collision avoidance with the environment from a global perspective to synthesize long-term human trajectories.

## 3 LOCO3D DATASET

### 3.1 OVERVIEW

Table 1 summarizes the statistics of existing human motion datasets and Loco3D, specifically, the number of frames, scene statistics, and subjects in the datasets. Our dataset contains 2500K frames of human trajectories in 130 scenes (see Figure 1 for examples). The number of scenes surpasses all the real human motion datasets. We collected multi-person interactions in semantically meaningful scenes, which is not included in most of the compared datasets. The number of trajectories for each person is 7071 in total. Furthermore, our dataset contains 3D human pose and mesh data in addition to trajectories. This makes Loco3D relevant for other scenarios, such as for use in AR/VR and for real-world human trajectory understanding.

Table 1: Statistics of existing human motion datasets and our proposed dataset.

| Dataset | Frame | Scene | | | | Subject | | | |
| | | Count | Semantics | Geometry | Location | Pos/Pose | Multi | Motion* | Target − action |
| --- | --- | --- | --- | --- | --- | --- | --- | --- | --- |
| HPS (Guzov et al., 2021) | 300K | 8 | | ✓(3D mesh) | Out/Indoor | 3D | ✓ | Real | Daily actions |
| EgoBody (Zhang et al., 2022) | 153K | 15 | | ✓(3D mesh) | Indoor | 3D | ✓ | Real | Daily actions |
| PROX (Hassan et al., 2019) | 100K | 12 | ✓ | ✓(3D mesh) | Indoor | 3D | | Real | Daily actions |
| GIMO (Zheng et al., 2022) | 129K | 19 | | ✓(3D mesh) | Indoor | 3D | | Real | Daily actions |
| Grand Station (Zhou et al., 2012) | 50K | 1 | | ✓(Aerial image) | Outdoor | 2D | ✓ | Real | Trajectory |
| SDD (Robicquet et al., 2016) | 929K | 6 | ✓ | ✓(Aerial image) | Outdoor | 2D | ✓ | Real | Trajectory |
| ETH (Ess et al., 2007) | 50K | 2 | | ✓(Aerial image) | Outdoor | 2D | ✓ | Real | Trajectory |
| THOR (Rudenko et al., 2020) | 360K | 3 | | ✓(3D point cloud) | Indoor | 2D | ✓ | Real | Trajectory |
| JRDB (Vendrow et al., 2023) | 636K | 30 | | ✓(3D point cloud) | Out/Indoor | 3D | ✓ | Real | Trajectory |
| GTA-IM (Cao et al., 2020) | 1000K | 10 | | ✓(3D mesh) | Indoor | 3D | | Synthetic | Trajectory |
| HUMANISE (Wang et al., 2022) | 1200K | 643 | ✓ | ✓(3D mesh) | Indoor | 3D | | Synthetic | Daily actions |
| CIRCLE (Araújo et al., 2023) | 4300K | 9 | ✓ | ✓(3D mesh) | Indoor | 3D | | Real | Daily actions |
| Loco3D (Ours) | 2500K | 130 | ✓ | ✓(3D mesh) | Indoor | 3D | ✓ | Real | Trajectory |

We use *"Real" to refer to natural motions and walking behaviors captured via video or motion capture, while "Synthethic" refers to animated motions and behaviors.

## 3.2 DATA COLLECTION

### 3.2.1 SYSTEM OVERVIEW

Figure 2 shows the overview of our data collection system. In our system, two people use VR headsets to enter a shared virtual environment, where they can interact with each other and are asked to perform tasks that require walking. The movements of the participants are captured in real time using a motion capture system. These movements are then mapped onto virtual avatars that move in the same way as the participants. This allows researchers to study how people move and interact in a virtual environment that mirrors the real world. Being able to see the other person's avatar also enables easy interactions while avoiding collisions. The advantages of our system are four-fold: (1) scene variation is fast and easy to achieve by editing the virtual scenes, such that human trajectory

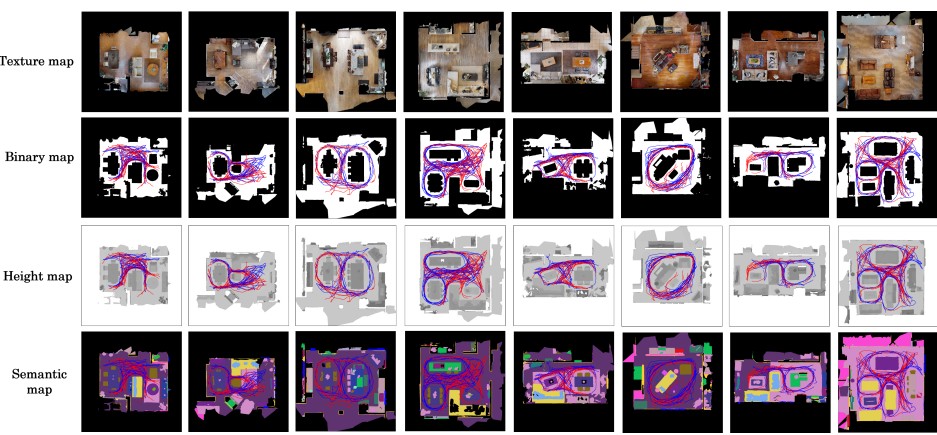

Figure 1: Loco3D contains 130 scenes with rich spatial information, such as photorealistic textures, 3D geometry, and semantics. Blue and red curves depict two people's trajectories for one data collection session.

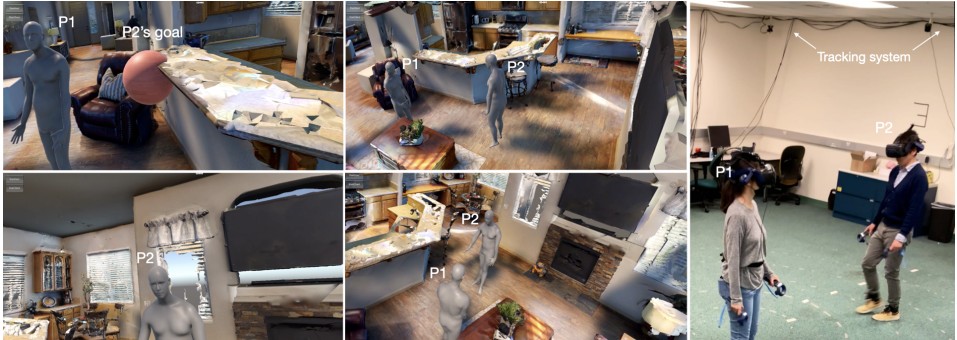

Figure 2: Overview of our data collection system in VR. Sets of two participants took part in the data collection process, wearing a VR headset, trackers, and hand controllers. They performed tasks that required walking in a physical space while reaching goals displayed in the virtual space, which matched the physical space in size. In VR, each participant embodies an avatar that matches their real-world behaviors in real time, to enhance awareness of each other's presence and movements. The motion tracking system and the VR system record the participants' trajectories and motions in real time. We use Habitat-matterport 3D dataset (Ramakrishnan et al., 2021) to create the virtual scenes.

data in a large variety of scenes can be collected; (2) accurate spatial information can be obtained by recording the spatial data in digital format; (3) the physical and the virtual spaces are of the same size, allowing the participants to walk naturally in the physical space and produce locomotion data in the virtual space; and (4) variables, such as scenes and social interactions between participants, can be accurately and easily controlled and replicated in a virtual space compared with a real-world setting.

### 3.2.2 EXPERIMENT

Experiments for the data collection were conducted with participants in an open space (10m by 10m). In the experiment, each person is assigned a unique goal, represented by a virtual marker that only they can see. Once a person reaches their goal, a new marker appears in a different location, starting a new round of the task. The new goal appears at a random location, which is at least 3m away from the previous goal position to prevent short trajectories. This process repeats around 5 minutes per scene to allow the participants to explore the virtual environment and add variations to the collected trajectory data. See Appendix A for more details of the data collection setup.

## 4 TRAJECTORY SYNTHESIS

### 4.1 OVERVIEW

We propose a UNet (Ronneberger et al., 2015) based pipeline that enables learning long-term trajectories considering scene geometry and multi-person interaction. Figure 3 shows our pipeline. It comprises two components: (1) a global path planner and (2) a local path planner. The global path planner generates a global path map, a static probability map of all possible path paths to the goal. The local path planner generates a dynamic trajectory to the goal based on the global path map. The stepwise approach from static global path to dynamic local path makes the model more stable for learning trajectories. Technical details of our model are introduced in the following sections.

### 4.2 GLOBAL PATH PLANNER

The inputs to the global path planner are the past trajectory of each participant (P1 and P2), the goal position of P1, and the scene map. These inputs are passed to the UNet to generate global path maps for two participants, $PM_{g,n}$, where $n$ is the person's index. A global path map is a probability map of each person's future trajectory without time-series variation. The global path maps for two participants are generated considering their interactions such that their trajectories do not overlap.

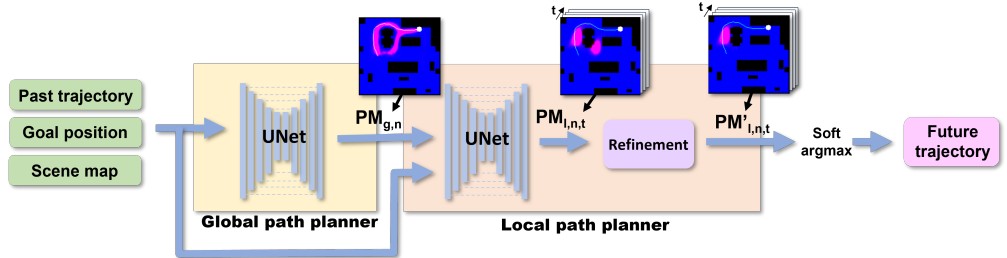

Figure 3: Overview of our method. It contains a global and a local path planner. The global path planner takes as input the past trajectory of two participants (P1 and P2), the goal position of P1, and the scene map, and produces a global path map as output. The global path map is further passed into the local path planner together with all the inputs to the global path planner. The local path planner generates a local path map, which is refined to acquire the predicted future trajectory.

### 4.3 LOCAL PATH PLANNER

The local path planner takes as input the global path map in addition to the inputs to the global path planner. First, UNet generates a local path map $PM_{l,n,t}$, which is a probability map of the person's time-series trajectory. The local path map is similar to the global path map, but it contains a time series variation of the probability map. With the local path map, a refinement process is applied to consolidate the peaks of the probability map $PM_{l,n,t}$ into a single location. Refined probability map $PM'_{l,n,t}$ is updated following equation 1 so that a single location can be given as a person's position. $PM_{l,n,t+1}$ represents the predicted local path map by the UNet, $PM'_{l,n,t}$ and $PM'_{l,n,t+1}$ denote the refined probability maps for the current time step $t$ and next time step $t + 1$. $\sigma$ is the variance of a Gaussian distribution and is set as 32 pixels.

$$PM'_{l,n,t+1} = \mathcal{N}(softargmax(PM'_{l,n,t} \cdot PM_{l,n,t+1}, \sigma)) \tag{1}$$

### 4.4 LOSS FUNCTION

Loss functions are defined in equation 2. $\mathcal{L}_g$ and $\mathcal{L}_l$ are loss functions for the networks in the global path planner and the local path planner. $N$ and $T$ represent the number of participants and time steps, respectively. $PM_g$ is the predicted global path map and $\hat{PM}_g$ is the ground-truth global path. $PM_l$ is the predicted local path map and $\hat{PM}_l$ is the ground-truth local path. Note that $\hat{PM}_g$ and $\hat{PM}_l$ are represented as a trajectory and a time series to the goal, respectively. We compute the sum of Binary Cross Entropy (BCE) (PyTorch, 2023) between $PM_{g,n}$ and $\hat{PM}_{g,n}$ as $\mathcal{L}_g$, and the sum of BCE between $PM_{l,n,t}$ and $\hat{PM}_{l,n,t}$ as $\mathcal{L}_l$. The total loss $\mathcal{L}_{total}$ is calculated as the sum of $\mathcal{L}_g$ and $\mathcal{L}_l$.

$$\mathcal{L}_g = \sum_{n=1}^{N} BCE(PM_{g,n}, \hat{PM}_{g,n}), \quad \mathcal{L}_l = \sum_{n=1}^{N} \sum_{t=0}^{T} BCE(PM_{l,n,t}, \hat{PM}_{l,n,t}), \quad \mathcal{L}_{total} = \mathcal{L}_g + \mathcal{L}_l \tag{2}$$

## 5 EVALUATION

### 5.1 EXPERIMENTAL SETUP

#### 5.1.1 PROBLEM DEFINITION AND EVALUATION METRIC

In this experiment, we aim to synthesize P1's trajectory considering the interaction between P1 and P2. The network's inputs are the past trajectory of both participants (15 frames each), the goal position of P1, and the scene map. The synthesis of P1's trajectory is conditioned on their goal but

not P2's goal. This is because we assume that P1 is unaware of P2's goals, and therefore, cannot use it as a basis for their path. Each person must navigate the environment independently to reach their own goal and deal with social situations, such as waiting for the other person before proceeding, as they arise. We evaluate the output, which is P1's future trajectory towards the goal (90 frames). The frame rate was downsampled to 15 fps from the original 90 fps to save memory.

We use binary maps as scene maps, with 1 indicating the area between -0.3m to 0.3m height above the floor level and 0 for all the other areas. The map size is 1024 pixels by 1024 pixels in the image and 10m by 10m in the physical world. All the training data are augmented by horizontal flipping and rotation with 90, 180, and 270 degrees, increasing the number of training data by eight-fold.

We use ADE (Average Displacement Error), a commonly used metric to evaluate the performance of trajectory synthesis. ADE refers to the mean squared error over all the estimated points and the ground-truth points on every trajectory. The ADE scale is represented in pixel units. Note that the value of ADE in pixels is approximately equivalent to ADE in cm in the physical world.

### 5.1.2 MODELS

**YNet (Mangalam et al., 2021)** YNet is one of the state-of-the-art techniques for goal-conditioned human trajectory prediction. It can generate long-term trajectories considering complex scene geometry with a UNet framework. We use YNet as a benchmark method for comparison.

**Ours** Our proposed model takes into account the presence of multiple people and their interactions when predicting trajectories. It reflects learned human trajectory more faithfully than YNet, as YNet introduces a heuristic process to improve performance stability. See Appendix D for details.

### 5.1.3 DATASETS

In the evaluation, the following three datasets were employed. Loco3D and GIMO for training, and Loco3D-R for testing though it is also used for training as a benchmark.

**Loco3D** Loco3D is our main contribution collected using our VR system. The dataset includes over 7000 multi-person trajectories in 130 indoor environments. We split it into training data (85%) and validation data (15%).

**Loco3D-R** To test the models on real-world data, we built a human trajectory dataset collected in a physical space. To collect this dataset, we had two participants walk around in a room that contained furniture. Each participant's movements were tracked and recorded by a motion capture system. They were then given a list of goals and asked to walk to each goal in sequence, one after another. This allowed us to record the full trajectories of their motion and social interactions necessary to navigate around the furniture and in 10 tight spaces. See Appendix B for details of Loco3D-R. In the case of using the dataset for both training and testing, it was divided following the cross-validation manner, specifically, all combinations of 8 scenes for training, 1 scene for validation, and 1 scene for testing.

**GIMO (Zheng et al., 2022)** GIMO is an indoor daily activity dataset containing locomotion data, and we employed GIMO as a benchmark dataset. GIMO contains a relatively large number of walking actions compared to other existing indoor datasets in real environments, while it has limited scene variations and only single-person data. We extracted the locomotion data and excluded trajectories that were too short ($< 2s$), resulting in 187 trajectories from 217 action sequences in 19 scenes. We divided the dataset into training (85%) and validation sets (15%).

### 5.2 RESULTS

### 5.2.1 QUANTITATIVE RESULTS

Table 2 reports the performance tested on Loco3D-R of YNet and our model trained with GIMO, Loco3D-R, and Loco3D. As can be seen, YNet and our model trained on Loco3D outperform other datasets. Multiple factors in Loco3D improve the performance of trajectory synthesis. It is clear that the large scale of the dataset is one of the dominant factors impacting model performance (see Appendix C.1). Additionally, multi-person interaction data contained in Loco3D is another likely

factor that helps performance improvement in multi-person scenes. To investigate the contribution of multi-person data, a benchmark test was performed as shown in Table 3. The evaluation compared the performance of our model with single-person and multiple-person trajectories. The results show that the trajectory synthesis performance using single-person trajectories is inferior to that using multi-person trajectories. This indicates that considering interaction between humans is a key component of synthesizing plausible human trajectories in multi-person environments.

Table 2: Benchmark of datasets. The table reports averaged ADEs [pixels] over three trials $\pm$ SD.

| $TrainData$ | $YNet$ | | | $Ours$ | | |
|---|---|---|---|---|---|---|
| | $t = 2s$ | $t = 4s$ | $t = 6s$ | $t = 2s$ | $t = 4s$ | $t = 6s$ |
| GIMO | $83.6 \pm 1.0$ | $105.9 \pm 4.4$ | $130.9 \pm 7.2$ | $87.3 \pm 18.3$ | $140.9 \pm 11.0$ | $173.9 \pm 4.8$ |
| Loco3D-R | $77.4 \pm 2.5$ | $91.3 \pm 2.7$ | $108.8 \pm 3.6$ | $56.2 \pm 2.5$ | $108.1 \pm 2.9$ | $150.8 \pm 3.1$ |
| Loco3D | $\mathbf{39.5 \pm 1.5}$ | $\mathbf{67.8 \pm 0.4}$ | $\mathbf{80.6 \pm 2.8}$ | $\mathbf{27.8 \pm 0.9}$ | $\mathbf{52.7 \pm 2.2}$ | $\mathbf{72.6 \pm 4.8}$ |

Table 3: Effect of multi-person data. The table reports averaged ADEs [pixels] over three trials $\pm$ SD.

| $Method + Dataset$ | $t = 2s$ | $t = 4s$ | $t = 6s$ |
|---|---|---|---|
| Ours + Loco3D (single-person) | $28.2 \pm 1.7$ | $55.2 \pm 2.6$ | $77.8 \pm 1.5$ |
| Ours + Loco3D (multi-person) | $\mathbf{27.8 \pm 0.9}$ | $\mathbf{52.7 \pm 2.2}$ | $\mathbf{72.6 \pm 4.8}$ |

### 5.2.2 QUALITATIVE RESULTS

Figure 4 shows a qualitative comparison of synthesized trajectories using our model trained on the three datasets. The results present the global path map and the local path map synthesized by our method. As can be seen, our model trained on Loco3D synthesizes a global path map clearly distributed on all the affordable paths to the goal; consequently, the local path map proceeds to the goal at each time step following the global path map. As a result, the estimated positions at each time step, shown as green circles, move towards the goal along the ground-truth trajectory. On the other hand, with Loco3D-R and GIMO, the global path map is distributed near the starting point or the goal, resulting in the estimated positions at each time step getting stuck around the starting point. This indicates that the model could not successfully learn possible trajectories because the dataset scales of Loco3D-R and GIMO are limited.

Figure 5 compares the synthesized trajectories by our model trained on single- and multi-person trajectory data. With multi-person data, the estimated paths of the two people do not overlap, similar to the ground-truth where the two people avoid collision during data collection. This is because the model simultaneously optimizes the paths for two people. On the other hand, with single-person data, the estimated path differs from the ground-truth because the model cannot take into account P2's motion. Consequently, P1's path prediction could lead to collisions, as seen in the trajectory overlaps.

## 6 LIMITATIONS AND FUTURE WORK

Our model considers the motion of other people to avoid collisions and overlapping paths. However, it does not synthesize socially aware trajectories that take into account complex human behaviors, such as politeness or personal space. Learning and synthesizing sophisticated multi-person interactions, such as "wait and go," is a future challenge to address. Since Loco3D contains a large number of trajectories with such social behaviors included, we consider such synthesis to be an interesting and challenging future work. Furthermore, Loco3D comprises rich spatial data, such as 3D geometry and scene semantics, 3D poses of all humans, and egocentric views with head IMU data; it can be utilized for various applications, including scene reconstruction from human motion data. In addition, extending the dataset to include various types of actions, such as multi-person cooperative tasks, could be one of the promising future directions. Additionally, there are gaps between virtual environments and the real world, such as a lack of physicality in VR objects and lower avatar expressivity compared

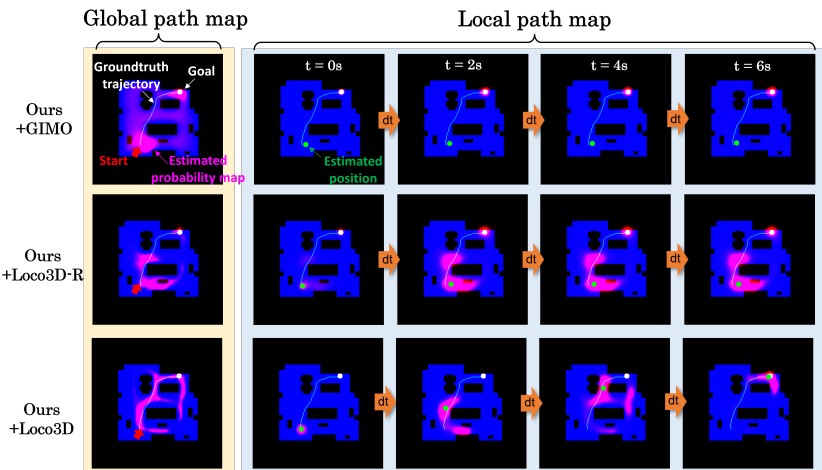

Figure 4: Trajectory synthesis results using our model trained on three datasets: GIMO, Loco3D-R, and Loco3D, seen from the top row to the bottom. The white paths are the ground-truth trajectories, and the purple curves present the probability maps predicted by our model. The green circles depict the estimated position on the trajectory at each time step (t=0.0s, 2.0s, 4.0s, and 6.0s). As can be seen, using Loco3D we could successfully predict the position of the person at each time step to match the ground-truth trajectory while with the other datasets, the predicted points at each time step are clustered near the starting position.

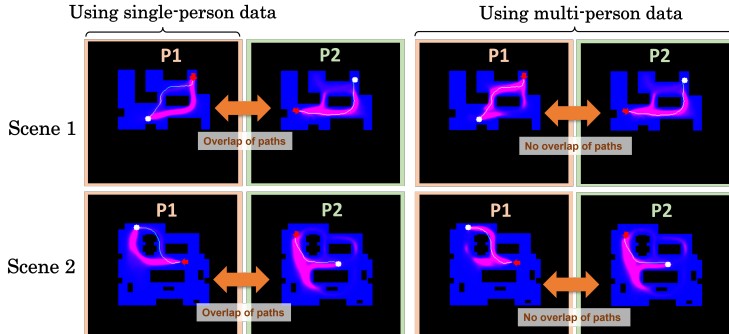

Figure 5: Trajectory synthesis results using our model trained on single- and multi-person trajectory data. Using multi-person data, our model predicts each participant's trajectory where collisions with the other participant are avoided, which occurs if we use only the single-person data.

to real people. They could affect behavioral details, such as walking speed or interpretation of non-verbal communication through facial expressions, in particular. The impact of these limitations and how well they compare to fully synthetic datasets are important to consider in future work.

## 7 CONCLUSION

We presented the Loco3D dataset, a novel dataset that focuses on human trajectory in complex indoor environments, containing multi-person interactions with full 3D body motions and highly accurate full spatial information in over 100 indoor scenes. In addition, we introduced a UNet-based trajectory synthesis method, enabling us to consider social interactions between humans and scene geometry from a global perspective to synthesize long-term trajectories. Experiments showed that, (1) the models trained with Loco3D outperform other indoor datasets evaluated on real-world test data, and (2) our model trained with Loco3D synthesizes socially aware trajectories in scenes with multi-person interaction, outperforming the state-of-the-art model. We hope this work can be used as a benchmark for future research on studying human motions and trajectories in indoor environments.

## REPRODUCIBILITY STATEMENT

We performed three trials on the training and testing process to ensure the reproducibility of the evaluations shown in the paper. Table 2 and 3 report averaged ADEs over three trials $\pm$ standard deviations. All the code and data used in our work can be found at `https://anonymous.4open.science/r/ICLR2024_anonymous-4118/README.md`

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

## A  VR SYSTEM FOR DATA COLLECTION

Details of the VR system used for the data collection are introduced in the following sections.

### A.1  SYSTEM STRUCTURE

The system receives real-time tracking data from the motion trackers worn by each participant. This data is used to update the avatars in the virtual environment so they accurately reflect the poses and movements of the participants. Each virtual avatar is represented by a SMPL mesh (Loper et al., 2015), calibrated to match the proportions of each person's body. To encourage movement and social interaction between the two participants, the system generates a goal object that each person needs to reach as part of the data collection task. This gives the participants a reason to move around and interact with each other in the virtual space, resulting in social behaviors, such as waiting for someone to pass before proceeding or backtracking for an oncoming person if the path is too narrow to allow them to cross.

### A.2  VR HARDWARE

Using the HTC VIVE system (VIVE, 2023), we track the movements of two people as they explore a virtual space. Each person wears a VR headset, holds a controller in each hand, and has three motion trackers (VIVE pucks) on their body - two on the ankles and one on the torso, for a total of six tracked points. This setup allows us to capture the full range of body movements as the participants interact with the virtual environment and with each other. The HTC VIVE's outside-in tracking system uses the six tracked points on each person's body to calculate their absolute pose and position with a high degree of accuracy. With a tracking frequency of 90 Hz and an accuracy within a few millimeters (Holzwarth et al., 2021), the system can track small movements and gestures in real time to provide a highly responsive and immersive experience, thereby eliciting natural walking and social behaviors, necessary for collecting realistic data.

### A.3  3D VIRTUAL ROOM DATA

We use the Habitat-matterport 3D dataset (Ramakrishnan et al., 2021) to create the virtual scenes. The dataset includes more than 1K 3D indoor spaces, which are captures of actual rooms using the Matterport 3D scanner (Matterport, 2023). Some of the scenes have semantic information added that Matterport has manually labeled. We used 130 scenes with full 3D scene geometry and semantic labels for collecting our data.

### A.4  ALIGNMENT OF VR AND THE REAL SPACE

The data collection was conducted in a VR lab (10m x 10m), which was larger than every virtual room we used. Firstly, we aligned the centers of the virtual and the physical rooms so that the virtual room was totally contained in the physical room. During data collection, goal positions were controlled to appear in the predefined virtual room to make the participants walk safely without getting close to the physical walls. If a participant got close to the physical walls, a virtual guardian appeared, indicating to the participant that they were too close to the boundary of the virtual space. This is a built-in safety feature of the VR headset we used.

### A.5  INSERTING ADDITIONAL VIRTUAL OBJECTS

Our data collecting system can allow flexible insertion of additional objects into the original 3D virtual rooms, e.g., an SMPL human sitting on the sofa and a trashcan placed on the floor (see Figure 6). Our experiments have not revealed how the virtual human and objects affect trajectory synthesis because the data covering these scenarios is limited. We plan to continue to extend the dataset by including more social interactions and mannerisms in our future work, making it richer with each new iteration.

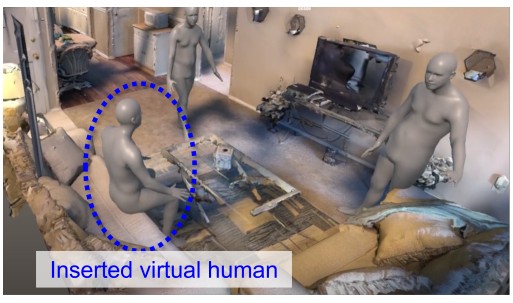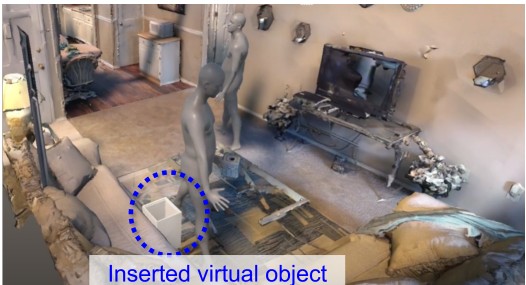

Figure 6: Inserted virtual objects.

# B  Loco3D-R: A dataset for test in the real world

Although Loco3D is collected in highly realistic virtual environments and useful for learning human trajectory considering the surrounding environment, it is a general concern that there might be a difference in human perception between the physical and virtual space that results in performance degradation when transferring from the virtual to the real world. To address the concern, we built Loco3D-R, a human trajectory dataset in the physical space, which can be used as test data to show that the model trained with Loco3D can be utilized in the real environment.

Collecting real-world human trajectory data was done in an empty room in a campus building. Two participants walked to conduct a task in the room where several pieces of furniture were placed, and their 3D motions and trajectories were captured by a motion capture system. The experiment was conducted in 10 different layouts with 5 participants, resulting in 1024 collected trajectories. Figure 7 illustrates the binary maps of the 10 scenes we collected in Loco3D-R.

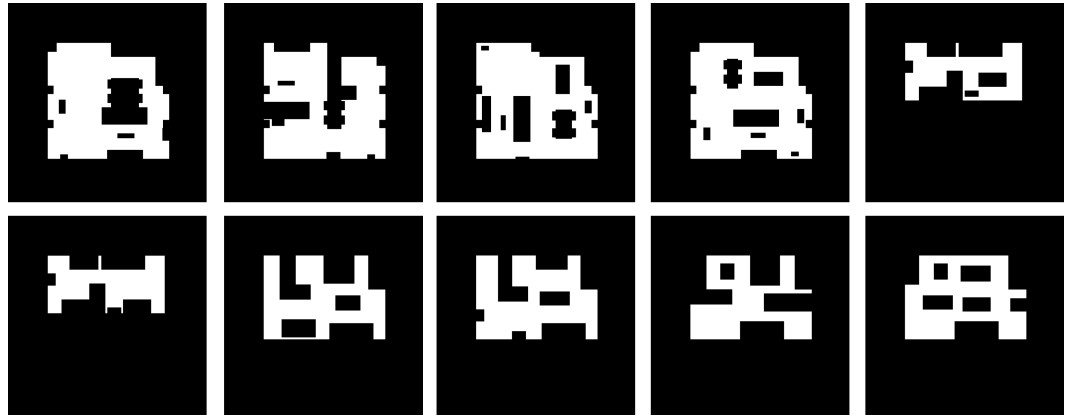

Figure 7: Binary maps of Loco3D-R dataset. We collected this dataset with 5 participants performing tasks in the shown 10 different layouts of a physical room.

# C  Additional experimental results

## C.1  Effect of dataset size

In this experiment, we study the effect of dataset size on the trajectory synthesis performance. Table 4 shows the results obtained by varying the number of scenes and trajectories in Loco3D to train our model. As can be seen, the dataset size significantly affects the trajectory synthesis performance tested on Loco3D-R. The trajectory synthesis performance using Loco3D is closest to Loco3D-R (#Trajectory=1048) when #Scene=20 (#Trajectory=969). This indicates the datasets collected in VR and the real world perform nearly equivalently with similar dataset sizes. With similar dataset sizes, we notice that training on Loco3D-R led to lower ADEs than training on Loco3D. This is

mainly because the training and testing scenes are similar in geometry when we use Loco3D-R for cross-validation evaluation. In contrast, the scene geometry is highly different and more complex in the VR scenes (see Figure 1 for the Loco3D scenes and Figure 7 for the Loco3D-R scenes). As the number of scenes and trajectories in Loco3D increases, as shown in Table 4, training on Loco3D achieved lower ADEs and better generalization performance tested on Loco3D-R.

Table 4: Effect of dataset size.

| $Method + Dataset$ | $\#Scene$ | $\#Trajectory$ | $ADE(Average Displacement Error)$ | | |
|---|---|---|---|---|---|
| | | | $t = 2s$ | $t = 4s$ | $t = 6s$ |
| Ours + Loco3D-R | 10 | 1048 | 56.2 | 108.1 | 150.8 |
| Ours + Loco3D | 10 | 362 | 81.6 | 131.7 | 167.2 |
| Ours + Loco3D | 20 | 969 | 68.3 | 118.1 | 159.8 |
| Ours + Loco3D | 50 | 2203 | 39.5 | 73.0 | 101.5 |
| Ours + Loco3D | 90 | 4310 | 29.3 | 55.9 | 78.3 |
| Ours + Loco3D | 130 | 5982 | 27.8 | 52.7 | 72.6 |

## C.2 Effect of scene map type

In this experiment, we investigate the effect of semantics and height information on the performance of trajectory synthesis. We used three types of maps: binary map, semantic map, and height map. The binary map has 1 for the area within -0.3m to 0.3m from the floor and 0 for all the other areas. The semantic map contains 40 semantic labels based on the NYU40 label (Silberman et al., 2012), provided by the Habitat-matterport 3D semantics dataset (Yadav et al., 2022). The height map contains the height information of each pixel. Height between -0.3 and 2m is normalized to 0 to 1. The evaluation was performed on Loco3D because Loco3D-R contains no semantics or height information of scenes. The Loco3D dataset was divided into training, validation, and testing with a ratio of 60%, 15%, and 25%, respectively.

Table 5 shows the performance with each map type. Both the semantic map and height map work better than the binary map, with the height map showing a significant improvement. The results indicate that human trajectory is determined not only by traversability but also by other factors in the scene, including semantics and 3D geometry. Based on this, it is expected that our dataset can be used to analyze the relationship between human behavior and environmental factors.

Additionally, the trajectory synthesis performance of our model tested on Loco3D using binary maps, as shown in Table 5, got higher ADEs compared with the performance tested on Loco3D-R, as shown in Table 4, though the number of trajectories in the test sets of these two setups is similar. This is mainly because the geometry of VR scenes is significantly more complex than that of the real scenes, e.g., complicated furniture layouts in VR (see Figure 1 for the Loco3D scenes and Figure 7 for the Loco3D-R scenes). Further, when tested on Loco3D, the number of training trajectories is 60% of all the trajectories in Loco3D, while we used 85% trajectories in Loco3D when testing the model on Loco3D-R. The different training dataset sizes could contribute to the performance discrepancy.

## C.3 Benchmark comparison with mathematical model

We conducted a benchmark experiment using a mathematical approach, ORCA (Van Den Berg et al., 2011), a trajectory synthesis method widely used in the robotics field. Given a goal and target speed, this method calculates an optimal trajectory towards the goal, avoiding collision with surrounding objects. We combined ORCA with a global path planning algorithm A* proposed by Hart et al. (1968) to support long-term trajectory synthesis. For a fair comparison with the data-driven methods, we assigned the target speed by averaging all the trajectory speeds in the Loco3D dataset.

Table 6 shows the performance of ORCA, YNet, and our model trained on Loco3D and tested on Loco3D-R. The trajectory synthesis performance with ORCA is inferior to YNet and our model.

Table 5: Benchmark of scene map types.

| Map | ADE (Average Displacement Error) | | |
|---|---|---|---|
| | $t = 2s$ | $t = 4s$ | $t = 6s$ |
| Binary map | 31.2 | 66.3 | 101.8 |
| Semantic map | 29.1 | 67.6 | 96.0 |
| Height map | **25.8** | **58.5** | **91.8** |

While YNet and our model learn a probabilistic decision in complex scenes from human behavior, ORCA only considers the efficiency of moving towards the goal, causing the performance difference.

Table 6: Benchmark of methods.

| Method | ADE (Average Displacement Error) | | |
|---|---|---|---|
| | $t = 2s$ | $t = 4s$ | $t = 6s$ |
| ORCA | 43.8 | 70.8 | 93.6 |
| YNet | 39.5 | 67.8 | 80.6 |
| Ours | **27.8** | **52.7** | **72.6** |

## C.4 HUMAN TRAJECTORY PREDICTION

While we focused on goal-conditioned trajectory synthesis in this paper, trajectory prediction, which is not goal-conditioned, is one of the most important tasks in the human locomotion research area. Therefore, we conducted an experiment to show the contribution of our dataset on trajectory prediction. We use our model and YNet by excluding goal position from the input for both training and testing. Table 7 shows the result of trajectory prediction. Similar to the goal-conditioned trajectory synthesis, the performance with Loco3D is better than the other two datasets for both models. This indicates that motion prediction is one of the potential applications of Loco3D.

Table 7: Benchmark of datasets on trajectory prediction. The table reports ADEs.

| TrainData | YNet | | | Ours | | |
|---|---|---|---|---|---|---|
| | $t = 2s$ | $t = 4s$ | $t = 6s$ | $t = 2s$ | $t = 4s$ | $t = 6s$ |
| GIMO | 73.0 | 119.9 | 159.0 | 60.6 | 117.4 | 160.9 |
| Loco3D-R | 88.2 | 122.4 | 153.2 | 54.4 | 102 | 135.3 |
| Loco3D | **44.6** | **83.4** | **116.8** | **36.8** | **75.4** | **103.2** |

## D IMPLEMENTATION DETAILS OF OUR MODEL

We use the Adam optimizer (Kingma & Ba, 2014) to train our trajectory synthesis model. The learning rate is 1.5e-4, and the batch size is 16. Each model is trained for up to 100 epochs on a single NVIDIA RTX 4080 graphics card with 8G memory, taking around 5 hours to train on Loco3D.

We use the same UNet structure as used by YNet (Mangalam et al., 2021) in our global and local path planners. Specifically, the global path planner takes as input a scene map, a goal position map, and the past trajectories of two participants for 15 frames. So the number of input channels is $1 + 1 + 2 \times 15 = 32$. The global path planner produces global path maps for both participants, resulting in 2 output channels. The local path planner takes the inputs to the global path planner and the two global path maps generated by the global path planner. Consequently, the number of input channels of the local path planner is $32 + 2 = 34$, and the number of output channels is 90 for the predicted future trajectory. The shape of each channel is 256 by 256 pixels.

In our model, time-series trajectory is handled in a multi-channel image format. Specifically, the 2D coordinate of a position on a trajectory is plotted on a blank image (256 by 256 pixels) with a Gaussian distribution, and the time-series data is contained in multi-channels. Similarly, the goal position is encoded as an image and concatenated with the multi-channel trajectory image when fed into the model.

Compared to YNet, there are two major differences. First, our model exploits the goal position as an input to the model to learn goal-conditioned trajectories, but YNet employs a non-goal-conditioned architecture. Consequently, the probabilistic paths synthesized by YNet are typically widely scattered to all possible destinations, and their refinement relies on a downstream process that guides the trajectory to the goal along heuristically determined waypoints. This heuristic process prevents YNet from potentially representing real human behavior. In addition, our model processes static trajectories in the global path planner and dynamic trajectories in the local path planner in stages, but YNet processes dynamic trajectories throughout the model. Since learning static trajectories is easier than learning dynamic trajectories due to the difference in dimension size, our stepwise approach makes the model more stable for learning trajectories.

The scene map was sampled from the 3D room dataset (HM3D) by manually cutting the predefined area and projecting the height map onto the aerial-view image that covers an area of 10m by 10m. The height map is then thresholded at 0.2m and converted into the binary map representing the walkable area.

## E    LOCO3D DATASET STATISTICS

In this section, we describe the statistics of the trajectory data contained in Loco3D. We collected trajectory data from 31 participants in total, resulting in 7071 trajectory sequences after data pre-processing. Since we collected trajectories from two participants simultaneously, each participant's trajectory was counted separately. We removed short trajectories (less than 2m or 2s) and poor motion tracking data in the data preprocessing phase.

Figure 8 shows the number of trajectories collected in each scene. The average number and standard deviation over 130 scenes are 54.0 and 32.0, respectively. The number of trajectories differs across scenes, resulting from the following factors. We collected a large number of trajectories in scenes where human interactions occur frequently (e.g., paths with a bottleneck). Also, the number of trajectories is affected by the speed preferences of the participants. Given the same amount of time, participants walking fast gave us more trajectories than participants who walked slowly. Further, the stability of the motion-tracking performance also affected the number of trajectories since trajectories with large tracking errors are removed in the data preprocessing.

Figure 9 shows the distance distributions of the trajectories. The figure shows that more than half of the trajectories are longer than 4m. Since virtual rooms are usually smaller than 7m by 7m, the distribution is reasonable to assume daily movements in a room.

Figure 10 shows the travel time distribution of the trajectories. It shows that more than half of the trajectories are longer than 5s, which would be enough to learn locomotion in a single room.

Figure 11 shows the minimum distance between two participants in each trajectory. As can be seen, about 25% of the trajectories are within 1m of the other participant, and more than 70% are within 2m (See Figure 12). It indicates that many of the trajectories could be influenced by the trajectories of the other participant when they are in close proximity, as people typically consider how their behaviors might affect others when they are located close to other people. In the rest of the cases where the participants were at least 2m away from each other, there could still be social interactions that involve passing through each other with a distance to respect other people's personal space or taking a less direct route to the goal to avoid the risk of physical conflict with the other.

## F    GAP BETWEEN VR AND THE REAL

We observed gaps between people's behaviors in the real scenes vs. the VR scenes. We speculate that the differences in behaviors could be attributed to the lower fidelity of SMPL avatars, which lack detailed facial expressions that can make interaction different from real-world interactions. The lack

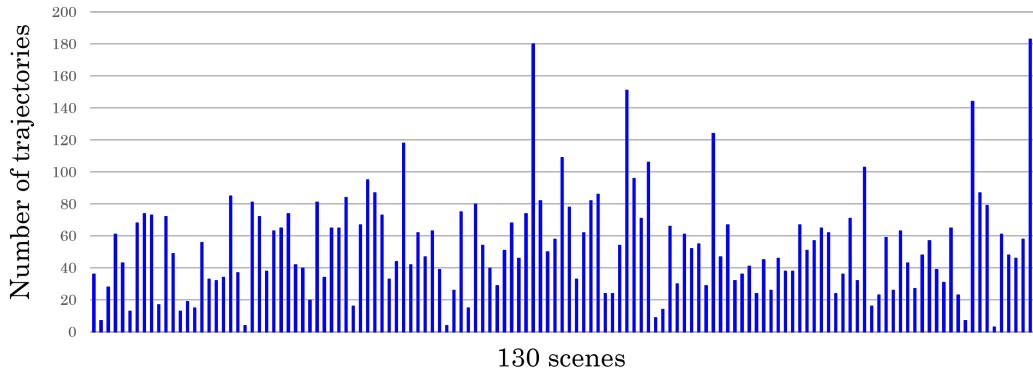

Figure 8: Number of trajectories in each scene.

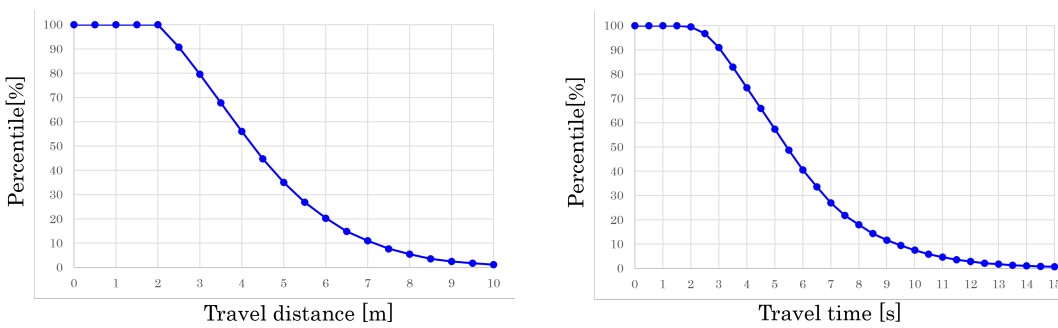

Figure 9: Travel distance of trajectories. Percentile represents the cumulative percentage of the travel distance that is above a certain value.

Figure 10: Travel time of trajectories. Percentile represents the cumulative percentage of the travel time that is above a certain value.

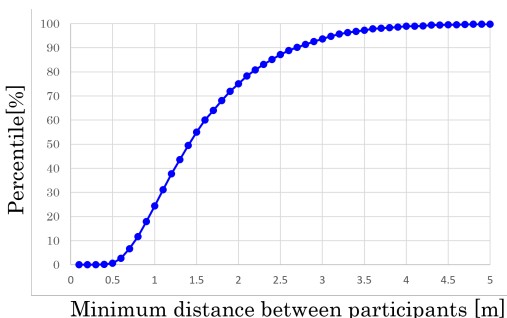

Figure 11: Minimum distance between two participants in each trajectory. Percentile represents the cumulative percentile of the minimum distance that is below a certain value.

of physicality of the objects can also impact user behavior as people may walk through furniture, which is impossible in the real world. However, this is less frequent as prior VR research has shown Simeone et al. (2017). Also, these gaps can be addressed by adding user interaction cues, such as adding colliders to objects or face trackers, and we expect that future advances in VR-UI would help bridge the gap between VR and the real world.

In addition, there is a visual difference between scan-reconstructed images observed in HM3D Ramakrishnan et al. (2021) and the images captured in the real world by cameras. There would be difficulty in using the scan-reconstructed images for other real-world applications, such as vision navigation of robots, due to the domain gap, however, we expect that the scan-reconstructed image

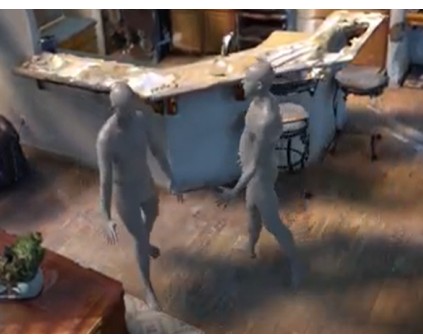

Figure 12: Scene of participants approaching each other

could be utilized for pre-training the model before fine-tuning on the real data, or utilized as the real images through domain adaptation James et al. (2019).

