# OpenReview forum: "Loco3D: Indoor Multiuser Locomotion 3D Dataset"
_ICLR.cc/2024/Conference — Submitted to ICLR 2024_

### Official Review · Reviewer_EhxJ · 2023-10-23

**Soundness:** 2 fair
**Presentation:** 3 good
**Contribution:** 2 fair
**Rating:** 5
**Confidence:** 5

**Summary:**

This paper presents a new multi-human-scene interaction dataset collected using a VR system with motion capture. This work also proposes a UNet-based model for human trajectory prediction and demonstrates its effectiveness on the proposed Loco3D dataset.

**Strengths:**

1. The idea of collecting human-scene interaction datasets using VR is great. It enjoys the benefit of real human behavior, scene diversity, and low cost (with scalability).
2. The Loco3D dataset seems a good contribution to the community and would interest multiple fields.
3. The idea of incorporating human-human interactions is well-motivated.

**Weaknesses:**

The main weakness of this work is that the current experimental analysis fails to align with the main characteristic of the dataset, making the motivation for creating the dataset less convincing.
- (motion) The dataset features locomotion (3D body motions), but the experiments are only on the trajectories.
- (scene-affordance) The dataset contains rich indoor 3D scenes with diverse objects and affordances, but the experiments contain only 'binary maps as scene maps' as scene representations, which cannot reflect the meaningful scene surroundings for human behaviors.
- (interaction) The motivation behind the dataset contains social interactions (e.g., social etiquette in section 1 paragraph 2), but the interactions in the experiments only involve collision avoidance and do not address the mentioned TV scenario.

While I believe the proposed dataset and the data collection pipeline would be interesting to the community, current experimental evaluations do not adequately reflect its contributions. Authors could consider more challenging tasks (human motion prediction/generation with complex scene conditions, like GIMO [a], CIRCLE [b], etc.). Or is there any specific difficulty in achieving this?

[a] Gaze-informed human motion prediction in context. ECCV 2022

[b] CIRCLE: Capture in Rich Contextual Environments. CVPR 2023

**Questions:**

- Could the author discuss more details on the VR mocap data collection protocol? e.g. the real and virtual space could be unaligned (e.g., a wall in the real world but not in the virtual and vice versa), how to mitigate the issue?
- More interaction types could be explored in the data collection process (e.g., two people need to accomplish certain tasks together).

Minor issues:
- Section 2.2 HUMANIZE -> HUMANISE
- [b] CIRCLE: Capture in Rich Contextual Environments needs to be cited and discussed.

---

> ### Author Response · Authors · 2023-11-22
> **Response to Reviewer EhxJ's comments**
>
> We thank you for your comments and feedback. In addition to the general updates, we address your concerns here.
>
> >The main weakness of this work is that the current experimental analysis fails to align with the main characteristic of the dataset, making the motivation for creating the dataset less convincing.
> (motion) The dataset features locomotion (3D body motions), but the experiments are only on the trajectories.
> We acknowledge that the 3D body motions are not yet fully used in our experiments, and incorporating the collected 3D data is a future direction we want to explore as described in Section 6 (Limitations and future work).
>
> (scene-affordance) The dataset contains rich indoor 3D scenes with diverse objects and affordances, but the experiments contain only 'binary maps as scene maps' as scene representations, which cannot reflect the meaningful scene surroundings for human behaviors.
> In addition to binary maps, we also evaluated the use of height maps and semantic maps to consider the spatiality and semantics of the indoor scenes when synthesizing trajectories. The semantic map contains 40 semantic labels based on the NYU40 label [4], provided by the HM3D dataset [5]. The height map contains the height information of each pixel. Height between -0.3 and 2m is normalized to 0 to 1. The experiment results are in Table 5. From the results, both the semantic map and height map work better than the binary map, with the height map showing a significant improvement. The results indicate that human trajectory is determined not only by traversability but also by other factors in the scene, including semantics and 3D geometry. Based on this finding, we expect that our 3D dataset can be used to analyze how environmental factors affect human behaviors.
> [4] Silberman et al. Indoor segmentation and support inference from RGBD images. 2012
> [5] Yadav et al. Habitat-Matterport 3D Semantics Dataset. 2022
>
> >(interaction) The motivation behind the dataset contains social interactions (e.g., social etiquette in section 1 paragraph 2), but the interactions in the experiments only involve collision avoidance and do not address the mentioned TV scenario.
>
>  In our current experiment, we focused mainly on the trajectory synthesis task by avoiding collision with furniture and the other person in an indoor environment. Since the global path planner uses overlapping avoidance as guidance for trajectory synthesis, the trajectories produced by our model can also take social distancing into account, i.e., overlapping paths are depreciated, otherwise people may walk too close to others. This is an example of the social mannerisms our current model addresses.
> Furthermore, our dataset contains scenarios where virtual humans, such as an SMPL human sitting on a sofa, are inserted into the 3D room scenes. However, we do not have experiment results on learning how the virtual human affected trajectory synthesis because the data covering these scenarios is limited. We plan to extend the dataset by including more social interactions in our future work. We show the demonstration of the inserted virtual human on our website https://sites.google.com/view/loco3d/home. Please refer to the video located at the bottom of the webpage from 0:30 to 0:49.
>
> >While I believe the proposed dataset and the data collection pipeline would be interesting to the community, current experimental evaluations do not adequately reflect its contributions. Authors could consider more challenging tasks (human motion prediction/generation with complex scene conditions, like GIMO [a], CIRCLE [b], etc.). Or is there any specific difficulty in achieving this?
> [a] Gaze-informed human motion prediction in context. ECCV 2022
> [b] CIRCLE: Capture in Rich Contextual Environments. CVPR 2023
>
>  We thank the reviewer for this suggestion. Predicting and synthesizing human motions is of interest to us. GIMO and CIRCLE mainly focus on motion generation with indoor scenes as conditions, and for our current work, we are interested in synthesizing plausible trajectories in indoor environments considering  social interactions between two people in an indoor environment. Although the focus is a bit different, we agree that motion generation could be coupled with trajectory prediction for many applications, such as designing home robots for task assistance. Given our current interest, we focused on collecting trajectory data and walking motions. Since GIMO and CIRCLE datasets contain more diverse motions including walking such as people interacting with objects in indoor scenes, as a next step we would like to explore how to use these datasets with our dataset to solve the task assistance problem, where locomotion and object interaction both need to be considered.

---

> ### Author Response · Authors · 2023-11-22
> **Response to Reviewer EhxJ's questions**
>
> >Could the author discuss more details on the VR mocap data collection protocol? e.g. the real and virtual space could be unaligned (e.g., a wall in the real world but not in the virtual and vice versa), how to mitigate the issue?
>
>  The data collection was conducted in a VR lab (10m x 10m) which was larger than every virtual room we used. Firstly, we aligned the centers of the virtual and the physical rooms so that the virtual room was totally contained in the physical room. During data collection, goal positions were controlled to appear in the predefined virtual room to make the participants walk safely without getting close to the physical walls. If a participant got close to the physical walls, a virtual guardian appeared indicating to the participant that they were too close to the boundary of the virtual space. This is a built-in safety feature of the VR headset we used. We have added these details to Appendix A.1.
>
> >More interaction types could be explored in the data collection process (e.g., two people need to accomplish certain tasks together).
>
>  We agree that more human interactions, as suggested by the reviewer, including various types of actions, is an interesting and promising topic. We plan to consider multi-person task completion as our next step and listed it as our in Section 6 (Limitations and future work).
>
> >Minor issues:
> Section 2.2 HUMANIZE -> HUMANISE
> [b] CIRCLE: Capture in Rich Contextual Environments needs to be cited and discussed.
>
>  We have corrected the typo in Section 2.2 and cited CIRCLE in Section 2.2 and Table 2 in Section 3.1.

---

### Official Review · Reviewer_Zavr · 2023-10-28

**Soundness:** 2 fair
**Presentation:** 3 good
**Contribution:** 3 good
**Rating:** 5
**Confidence:** 3

**Summary:**

The paper introduces a human behavior data collection system that utilizes VR to get a multi-person trajectories dataset, Loco3D, across 130 complex indoor settings. Additionally, the authors propose a human trajectory prediction model consider the multi-person scenario. Experimental outcomes indicate that in multi-person scenarios, both the Loco3D dataset and the proposed methods enhance trajectory synthesis outcomes.

**Strengths:**

* Leveraging VR to collect the multi-person trajectory is a compelling approach, considering the time cost and complexity to set up cameras or environments in the real world. The advantage is that although the scene is some scan-reconstructed, the human trajectory is real.
* The Loco3D dataset includes much more scenes then previous multi-person real dataset. The high diversity in the layouts can support more work focusing on trajectory synthesis in multi-person scenario.
* Their experiments demonstrate that the collected data can be used to improve the performance of the models, and the scale of the data is important.
* Their methodology takes into account multi-person trajectories, yielding enhanced results in comparison to prior research.

**Weaknesses:**

* Regarding the dataset statistics, there's an absence of comparisons concerning the number of trajectories in each scene, as well as their length and complexity. In Table 2, prior multi-person trajectory datasets, such as JRDB, contained approximately 20K frames for each scene. In contrast, Loco3D offers only 7.7K frames. It remains ambiguous whether this frame count pertains to a single trajectory or multiple ones. Additionally, the variation in the number of individuals across datasets is not clearly demonstrated.
* For the comparison with prior dataset, the most related prior dataset shold be the JRDB ones, which also contains the multi-person data. Current comparison is hard to see if the improvement is from the data scale or from different task settings.
* For the proposed method, the structure seems that it can only work for a fixed number of people. This limit the generality the proposed methods. It’s also hard to see if the proposed method can still be adapted to the single-person scenario and what the performance will be.
* For the qualitative results, treating the overlapping of the trajectory as a judgement is not proper. The trajectory also involves the time, two people may not go to a near position at each time step even if their trajectories overlap.

**Questions:**

* For the Loco3D dataset, does it also include the first-person view frames? Then how to deal with the gap between the rendered images using a scan-reconstructed dataset (HM3D) and the real world?
* The motivation mentions the human trajectory should consider if the other human is watching TV. However, based on the paper data collection process, all humans can only walk around, lack of the diversity in different social scenario. So what’s the actual social constraints covered in the dataset, instead of only collision avoidance between people?
* Does the dataset only contains two people scenario and why the design is like this?
* For the proposed method, is there some solution to make it adapt to scenarios with different number of people?
* Why not comparing the results with JBDR to see if the improvement really comes from different scenes or just different number of trajectories?

---

> ### Author Response · Authors · 2023-11-22
> **Response to Reviewer Zavr's comments**
>
> We thank you for your comments and feedback. In addition to the general updates, we address your concerns here.
>
> >Regarding the dataset statistics, there's an absence of comparisons concerning the number of trajectories in each scene, as well as their length and complexity. In Table 2, prior multi-person trajectory datasets, such as JRDB, contained approximately 20K frames for each scene. In contrast, Loco3D offers only 7.7K frames. It remains ambiguous whether this frame count pertains to a single trajectory or multiple ones. Additionally, the variation in the number of ...
>
>  We have added a description of our dataset statistics to Appendix E, specifying the total number of trajectories, the number of trajectories in each scene, the length of trajectories, and the travel time of trajectories. The number of trajectories is 7071 in total, and the number of trajectories for each person is counted separately (modified Section 3.1). Also, our dataset has nearly 20K frames on average for each scene, which is equivalent to JRDB. However, the number of frames is dependent on the sensor frame rate, so it is not necessarily meaningful to evaluate the dataset's scalability. For example, datasets captured by different sensors may have the same number of frames, however, it does not necessarily mean the number of trajectories or the length of trajectories are the same if the frame rates are different for these sensors.
>
> >For the comparison with prior dataset, the most related prior dataset shold be the JRDB ones, which also contains the multi-person data. Current comparison is hard to see if the improvement is from the data scale or from different task settings.
>
>  We found that the dominant factor that contributed to the performance improvement using our dataset is the dataset scale, based on the experiment discussed in Appendix C.1.
> As for JRDB, while it contains multiple people in indoor scenes, their focus is on a group of people or crowd behavior on a campus field or in campus buildings, which is similar to crowded open outdoor datasets such as the Grand Station Dataset [3]. In contrast, our focus is on the person-to-person interactions (e.g. waiting for others to go first, taking a less direct route to avoid colliding with others, etc) in small-scale home scenes. Additionally, the geometry complexity in indoor environments is highly different from that in open spaces, leading to different human behaviors. Due to the differences in space size and environment geometry, directly using the datasets collected in open spaces, including the JRDB ones, couldn’t allow models to successfully synthesize social interactions in indoor scenes. To our knowledge, there is no public dataset available containing multi-person trajectories and motions collected in home environments. We believe building such a dataset is necessary to support future research in this domain, as considering human interactions when synthesizing and predicting trajectories is important for various applications, like designing home robots for task assistance.
>
> [3] Zhou et al. Understanding collective crowd behaviors: Learning a mixture model of dynamic pedestrian-agents. 2012
>
> >For the proposed method, the structure seems that it can only work for a fixed number of people. This limit the generality the proposed methods. It’s also hard to see if the proposed method can still be adapted to the single-person scenario and what the performance will be.
>
>  We acknowledge that our model does not support dynamically changing the number of people. However, if we pre-define the number of people, our model can be adapted to the single-person or multi-person scenarios. The performance in the single-person scenario is in Table 3, where our model exploits only single-person data as the input to synthesize trajectories (ignoring the other person’s data). In this case, the performance is still comparable to the two-person scenario.
>
> >For the qualitative results, treating the overlapping of the trajectory as a judgement is not proper. The trajectory also involves the time, two people may not go to a near position at each time step even if their trajectories overlap.
>
>  The reason we mentioned overlapping as a judgment is mainly because the ground-truth trajectories (white curves) do not overlap. We acknowledge that the overlapping of trajectories does not necessarily mean collision with other people considering time. However, humans may prefer to choose trajectories, when possible, without overlapping with others regardless of time because they take into account not only collision avoidance but also social distancing, as we observed in our data collection sessions. Especially in indoor home environments due to the limited space, the social distancing tendency is even more significant than in large open spaces regarding chosen trajectories. This observation motivated the design of our model to avoid overlapping by the global path planner for the trajectory synthesis task.

---

> ### Author Response · Authors · 2023-11-22
> **Response to Reviewer Zavr's questions**
>
> >For the Loco3D dataset, does it also include the first-person view frames? Then how to deal with the gap between the rendered images using a scan-reconstructed dataset (HM3D) and the real world?
>
>  We can extract the first-person view image by taking a snapshot of the HM3D virtual room from the VR headset position and pose. The first-person data could be used for additional tasks in scan-reconstructed environments, but there would be difficulty in using it for other real-world applications, such as vision navigation of robots, due to the domain gap between scan-reconstructed and real images. However, we expect that the scan-reconstructed image could be utilized for pre-training the model before fine-tuning on the real data, or utilized as the real images through domain adaptation [3].
>
> [3] James, et al. Sim-to-real via sim-to-sim: Data-efficient robotic grasping via randomized-to-canonical adaptation networks, 2019
>
> >The motivation mentions the human trajectory should consider if the other human is watching TV. However, based on the paper data collection process, all humans can only walk around, lack of the diversity in different social scenario. So what’s the actual social constraints covered in the dataset, instead of only collision avoidance between people?
>
>  Our dataset contains additional objects inserted into the 3D virtual rooms, e.g., an SMPL human sitting on the sofa, and a trashcan placed on the floor. However, we do not have experiment results on learning how the virtual human and objects affect trajectory synthesis because the data covering these scenarios is limited. We plan to continue to extend the dataset by including more social interactions and mannerisms in our future work, making it richer with each new iteration. The discussion has been added in Appendix A.5
> We show the demonstration of the inserted virtual human and objects on our website https://sites.google.com/view/loco3d/home (anonymous during the review process). See the video at the bottom of the website from 0:30 to 0:49.
>
> >Does the dataset only contains two people scenario and why the design is like this?
>
>  Loco3D contains only two-person scenarios because our focus is on synthesizing trajectories in small-scale home spaces and not in crowded open spaces as other existing datasets such as JRDB. In such home environments, a two-person scenario can be considered commonplace. We think extending the setup to include more people or robot agents in future work would be a worthwhile direction to pursue.
>
> >For the proposed method, is there some solution to make it adapt to scenarios with different number of people?
>
>  Our model could be adapted to different numbers of people scenarios by pre-defining the number. At present, the trajectory synthesis performance in more than two people scenarios is unknown because we designed our dataset and model to work with two-person scenarios in home environments.
>
> >Why not comparing the results with JBDR to see if the improvement really comes from different scenes or just different number of trajectories?
>
>  We acknowledge that separating the evaluation of the number of trajectories and scenes would help clarify the contribution of our dataset. However, we think comparing with JRDB might not help answer if the improvement came from the number of scenes or trajectories because it would be an unfair comparison due to the highly different scene geometry, scale, and number of people. Specifically, JRDB contains the behavior of a group of people on a campus field or in campus buildings, which is similar to crowded open outdoor datasets. In contrast, we focus on two-person interactions in small-scale home scenes. In addition to the difference of space size and number of people, the geometry complexity in indoor environments is highly different from that in open spaces, leading to distinct human behaviors. More detailed description on the difference between JRDB and our dataset is in the response to the previous comment on JRDB.

---

### Official Review · Reviewer_v7u4 · 2023-10-31

**Soundness:** 3 good
**Presentation:** 3 good
**Contribution:** 2 fair
**Rating:** 3
**Confidence:** 3

**Summary:**

This manuscript presents Loco3D, a dataset of a pair of humans interacting with high-resolution indoor scenes in VR that includes detailed 3D body pose as well as detailed maps of the indoor environments. The dataset includes 7000 example trajectories across 130 scenes, and in addition to 3D body keypoints they provide semantic scene segmentation and scenes with photorealistic textures. They develop a UNet based path planner module that uses a path history, the goal location, and scene map to produce a probability map of trajectories. They consider three evaluation datasets – Loco3D, Loco3D-R which was collected in the real world, adn GIMO, a previously published dataset. They show improved performance compared to YNet on the Loco3D, but not GIMO datasets, and that training on Loco3D produced superior results. They show training with multi-person data is superior and give qualitative examples.

**Strengths:**

* The dataset is a contribution to the field and has several novel elements, including multi-person data, photorealistic textures, and semantic segmentation. I can see this experimental approach for generating training data to become common int he field. There is also a real-world test example.

* There is a new U-Net based modeling format that incorporated multi-person data and evaluations show some modeling improvements with multi-person data.

* The supplement and text are comprehensive and describe experiments well.

**Weaknesses:**

* The contributions can be distinguished from other datasets and models for human trajectory synthesis but the advance seems somewhat incremental in comparison. In particular the contribution is more the dataset than the model and so I wonder whether ICLR is the right venue. Because the dataset does not open up a new field in learning representations, but more advances the existing field it may find a better home in a more specialized venue.

* The distinctions between the modeling component and existing literature are unclear. The approach seems novel but also related to approaches like YNet and the strengths and weaknesses could be more clearly elucidated in the text. Moreover I would like to see benchmarks with other approaches to improve the contribution of the new models, even if this means computing on single person trajectories alone.

* There is not a robust comparison across standard benchmarks of the modeling component. It would be nice to know whether their proposed algorithm is SOTA and comparing its performance on a standard benchmark or whether the increase in performance is specific to the collected datasets. In fact the poor performance on GIMO is a limitation of the work in my opinion rather than just an endorsement of the value of the corpus.

**Questions:**

* Table 1 category of ‘real/synthetic’ is a bit ambiguous here, since the scenes are synthetically rendered.

* Units in Table 2?

* It is unclear how to interpret the poor performance on other datasets in Figure 4 and Table 2. The planner does not appear to work very well and it is unclear if this is just a domain gap? Moreover I was expecting to see comparisons training on the Loco3D corpus and testing on Loco3D-R

* Can YNet be extended to include multi-person trajectories?

* Can you comment on domain gap with real scenes. Unclear how human interaction in freely moving VR different from real environments. Affects the scope and generality of the method.

---

> ### Author Response · Authors · 2023-11-22
> **Response to Reviewer v7u4's comments**
>
> We thank you for your comments and feedback. In addition to the general updates, we address your concerns here.
>
> >The contributions can be distinguished from other datasets and models for human trajectory synthesis but the advance seems somewhat incremental in comparison. In particular the contribution is more the dataset than the model and so I wonder whether ICLR is the right venue. Because the dataset does not open up a new field in learning representations, but more advances the existing field it may find a better home in a more specialized venue.
>
>  We believe that our dataset has the potential to help develop learning representations to model human interactions in complex indoor geometries. Examples include taking a less direct route to avoid colliding with others or yielding to others.  To the best of our knowledge, we have not seen such interactions in publically available datasets. In our submission, while we emphasized the dataset contribution more than the algorithm similar to [1] presented in ICLR2023, we believe the experiment results can show the effectiveness of using our dataset to solve the important trajectory synthesis problem and open up new opportunities for considering social mannerisms in this problem for future research. Our experiments may serve as benchmarks for future researchers to build novel methods in this domain.
>
> [2] Li et al. A minimalist dataset for systematic generalization of perception, syntax, and semantics. ICLR 2023
>
> > The distinctions between the modeling component and existing literature are unclear. The approach seems novel but also related to approaches like YNet and the strengths and weaknesses could be more clearly elucidated in the text.
>
>  We have added descriptions of the specific differences between our model and YNet to Appendix D to clarify. To summarize the differences, our model exploits the goal position as an input to the model to learn goal-conditioned trajectories, but YNet employs a non-goal-conditioned architecture. Consequently, the probabilistic paths synthesized by YNet are typically widely scattered to all possible destinations, and their refinement relies on a downstream process that guides the trajectory to the goal along heuristically determined waypoints. This heuristic process prevents YNet from potentially representing real human behavior.
> In addition, our model processes static trajectories in the global path planner and dynamic trajectories in the local path planner in stages, but YNet processes dynamic trajectories throughout the model. Since learning static trajectories is easier than learning dynamic trajectories due to the difference in dimension size, our stepwise approach makes the model more stable for learning trajectories.
>
> > Moreover I would like to see benchmarks with other approaches to improve the contribution of the new models, even if this means computing on single person trajectories alone.
> There is not a robust comparison across standard benchmarks of the modeling component. It would be nice to know whether their proposed algorithm is SOTA and comparing its performance on a standard benchmark or whether the increase in performance is specific to the collected datasets. In fact the poor performance on GIMO is a limitation of the work in my opinion rather than just an endorsement of the value of the corpus.
>
>  Similar to prior work introducing new datasets [2], our focus is not on the model but on the dataset. Therefore, we don’t present any benchmarks to highlight the performance of our model in the paper. Instead, we demonstrate the value of our dataset in the experiments section by using the model we designed on pre-existing datasets to show a comparison of performance.
> While it may seem that poor performance on GIMO is a limitation of our work, in fact, the performance difference comes from the relatively limited scale of the trajectory contained in GIMO (less than 200 trajectories). Table 4 in Appendix C.1 shows how scalability affects performance.
> We want to highlight the reason why we focused on presenting the dataset contribution in this paper. Existing indoor datasets have limited scalability because collecting large-scale datasets in the real world, indoor rooms in our case, is very time-consuming. This is the reason that motivated us to exploit a VR system to collect scalable data, and we believe our data collection setup would be helpful for future work to address similar problems caused by the scalability of data.

---

> ### Author Response · Authors · 2023-11-22
> **Response to Reviewer v7u4's questions**
>
> >Table 1 category of ‘real/synthetic’ is a bit ambiguous here, since the scenes are synthetically rendered.
>
>  In Table 1, we have added a footnote* to the Motion column to clarify that “Real” refers to natural human motions and walking behaviors captured via video or mocap or any other means, while “Synthetic” refers to animated motions. Additionally, we have added further discussion regarding the gap between capturing human motion and behaviors in real scenes vs. virtual reality scenes that may affect human behavior in Section 6 (Limitations and Future Work) and Appendix.F. We observed gaps between people’s behaviors in the real scenes vs. the VR scenes. We speculate that the differences in behaviors could be attributed to the lower fidelity of SMPL avatars which lack detailed facial expressions that can make interaction different from real world interactions.
>
> > Units in Table 2?
>
>  All the units on ADEs are with pixels on the image (described in Section 5.1.1). We have added the units to each table caption to clarify.
>
> >It is unclear how to interpret the poor performance on other datasets in Figure 4 and Table 2. The planner does not appear to work very well and it is unclear if this is just a domain gap?
>
>  The relatively poor performance of GIMO and Loco3D-R is mainly due to the dataset size. The experimental results in Table 4 (Appendix C.1) show that the dominant factor of the performance difference between Loco3D and Loco3D-R is the dataset size. Similarly, we can infer that the performance gap between GIMO and Loco3D is caused by the dataset size, since the size of GIMO is much smaller than that of Loco3D. In addition, the inclusion of multi-person data in our dataset also improves the trajectory synthesis performance, as shown in Table 3 (Section 5.2.1).
>
> >Moreover I was expecting to see comparisons training on the Loco3D corpus and testing on Loco3D-R
>
>  In Table 2 (Section 5.2.1), all the results are obtained from training on Loco3D and testing on Loco3D-R. For your reference, we explained the training and testing datasets in Section 5.1.3.
>
> > Can YNet be extended to include multi-person trajectories?
>
>  We tried to extend YNet to include multi-person trajectories but obtained unsatisfying results. Specifically, YNet failed to synthesize trajectories for more than one person with no overlapping with each other. The reason is that YNet synthesizes probable trajectories widely distributed mainly due to its non-goal-conditioned architecture, and its refinement relies on a downstream process that guides the trajectories to the goal along heuristically determined waypoints. This heuristic refinement process prevents YNet from successfully synthesizing multi-person trajectories due to the lack of a global view when determining the waypoints to avoid overlapping with other people’s trajectories. The discussion has been added to Appendix.D.
>
> > Can you comment on domain gap with real scenes. Unclear how human interaction in freely moving VR different from real environments. Affects the scope and generality of the method.
>
>  We have added further discussion regarding the gap between capturing human motion and behaviors in real scenes vs. virtual reality scenes that may affect human behavior in Section 6 (Limitations and Future Work) and Appendix.F. We observed gaps between people’s behaviors in the real scenes vs. the VR scenes. We speculate that the differences in behaviors could be attributed to lower fidelity of SMPL avatars which lack detailed facial expressions that can make interaction different from real world interactions. The lack of physicality of the objects can also impact user behavior as people may walk through furniture, something that is not possible in the real world. However, these gaps can be addressed by adding user interaction cues, such as adding colliders to objects or face trackers, moreover, we expect that future advancements in VR-UI would help bridge the gap between VR and the real world.

---

### Official Review · Reviewer_Yanw · 2023-10-31

**Soundness:** 3 good
**Presentation:** 2 fair
**Contribution:** 3 good
**Rating:** 5
**Confidence:** 2

**Summary:**

The paper proposes a novel indoor human motion forecasting dataset containing paired motion of two real persons in virtual environments. To address the proposed task of socially-aware trajectory forecasting the authors further propose a U-Net-style model for socially-aware trajectory forecasting.

**Strengths:**

Social interactions in 3D scenes is highly relevant but under-explored. The authors approach of utilizing VR to easily generated large variations of virtual worlds is clever.

**Weaknesses:**

The authors over-claim their contributions by saying that their dataset represents “real” social interactions: a better description would be “hybrid” or “mixed” as the scene is entirely virtual. Also, real social interactions require humans to see each others faces - for observing small social cues - which is not possible with VR headset. The authors should adjust the description of their method as “real” in Table 1 and tone down their claims of representing real social interactions.

There are two concerns with regards to the proposed U-Net:
First,  the U-Net in Section 4 is not well-described:
* How is the scene sampled into an image?
* How is the heat map generated?
* How are past trajectories encoded?
* How is the goal encoded?
* How is the map encoded?
Second, the authors should have shown the effectiveness of their method on the experiments proposed in YNet.

**Questions:**

* What is part of the dataset? Will the authors make available the SMPL parameters at each frame as well?
* How does the speed of the person behaves after forecasting? Do they slow down when approaching the target? Is there a velocity “jump” when changing from past to future motion?
* Why is FDE not used in the experiments?
* For completeness: The dataset contains personal information of the recorded subjects: did the subjects consent to the release of their trajectory and pose data?
* For completeness: where will the dataset be made available?

**Details Of Ethics Concerns:**

The recorded dataset contains human motion which could be linked to individuals.

---

> ### Author Response · Authors · 2023-11-22
> **Response to Reviewer Yanw's comments**
>
> We thank you for your comments and feedback. In addition to the general updates, we address your concerns here.
>
> > The authors over-claim their contributions by saying that their dataset represents “real” social interactions: a better description would be “hybrid” or “mixed” as the scene is entirely virtual. Also, real social interactions require humans to see each others faces - for observing small social cues - which is not possible with VR headset. The authors should adjust the description of their method as “real” in Table 1 and tone down their claims of representing real social interactions.
>
>  In Table 1, we have added a footnote* to the Motion column to clarify that “Real” refers to natural human motions and walking behaviors captured via video or mocap or any other means, while “Synthetic” refers to animated motions. Additionally, we have added further discussion regarding the gap between capturing human motion and behaviors in real scenes vs. virtual reality scenes that may affect human behavior in Section 6 (Limitations and Future Work) and Appendix.F. We observed gaps between people’s behaviors in the real scenes vs. the VR scenes. We speculate that the differences in behaviors could be attributed to the lower fidelity of SMPL avatars which lack detailed facial expressions that can make interaction different from real world interactions. The lack of physicality of the objects can also impact user behavior as people may walk through furniture, something that is not possible in the real world. However, this is less frequent as prior VR research has shown[1]. Also, these gaps can be addressed by adding user interaction cues, such as adding colliders to objects or face trackers, and we expect that future advances in VR-UI would help bridge the gap between VR and the real world.
>
> [1]Simeone.A et.al. Altering user movement behaviour in virtual environments. IEEE transactions on visualization and computer graphics 2017
>
> > There are two concerns with regards to the proposed U-Net: First, the U-Net in Section 4 is not well-described:
> How is the scene sampled into an image?
> How is the heat map generated? ...
>
>  A detailed description of the U-Net used in our model is in Appendix D. In this section, we have included additional technical details to answer the questions regarding the preprocessing and encoding of the trajectory, scene map, and goal.
> In our model, time-series trajectory is handled in a multi-channel image format. Specifically, the 2D coordinate of a position on a trajectory is plotted on a blank image (256 by 256 pixels) with a Gaussian distribution, and the time-series data is contained in multi-channels. Similarly, the goal position is encoded as an image, and it is concatenated with the multi-channel trajectory image when fed into the model. The scene map was sampled from the 3D room dataset (HM3D) by manually cutting the pre-defined area and projecting the height map onto the aerial-view image that covers a 10m by 10m area. Then, the height map is thresholded at 0.2m and converted to the binary map representing the walkable area.
> In addition to the binary map, we conducted experiments using height and semantic maps to study the effect of spatiality and semantics of indoor scenes when synthesizing trajectories. Please find the processing setup of these two types of maps in Appendix C.2.
>
> > Second, the authors should have shown the effectiveness of their method on the experiments proposed in YNet.
>
> Similar to prior work introducing new datasets, our focus is not on the model but on the dataset. Therefore, we don’t present any benchmarks to highlight the model performance in the paper. Instead, we demonstrate the value of our dataset in the experiments section by using the model we designed on pre-existing datasets to show a comparison of performance.

---

> ### Author Response · Authors · 2023-11-22
> **Response to Reviewer Yanw's questions**
>
> > What is part of the dataset? Will the authors make available the SMPL parameters at each frame as well?
>
>  We plan to publish our dataset containing the 3D position/pose of each motion tracker (head, hands, waist, feet) for each person, as well as estimated SMPL parameters at each frame.
>
> > How does the speed of the person behaves after forecasting? Do they slow down when approaching the target? Is there a velocity “jump” when changing from past to future motion?
>
>  We interpret these questions to be asking 1) if the speed reaches zero at the end of the synthesized trajectory, 2) if the synthesized speed slows down when people get close to their targets, and 3) if the speed changes suddenly from zero to a positive value from the end of the past trajectory to the beginning of the synthesized trajectory.
> To answer these questions, our model was able to learn speed variations from a given past trajectory to the goal, so the model can synthesize the speed that smoothly transits from zero at the end of the past trajectory to a mean speed learned from the training data and slows down when people are approaching their goals and finally becomes zero after reaching the goals.
>
> > Why is FDE not used in the experiments?
>
>  We decided not to employ FDE as the evaluation metric since our method assumes the goal position of P1 is known (see Section 5.1.1).
>
> > For completeness: The dataset contains personal information of the recorded subjects: did the subjects consent to the release of their trajectory and pose data?
>
>  We conducted the human subject experiment with approval from the IRB at our institution (protocol #anonymized). All participants provided informed consent for both the data collection phase and the public release of the collected data.
>
> > For completeness: where will the dataset be made available?
>
>  Our dataset will be published on GitHub and linked from the following (currently anonymous) webpage after the review process: https://sites.google.com/view/loco3d/home

---

### Author Response · Authors · 2023-11-22

Dear reviewers,
We appreciate you for taking your precious time to review our paper and providing valuable comments. We have taken your insightful comments to improve our paper. We have carefully considered the comments and questions and tried our best to address every one of them. The authors welcome any further constructive comments. Below we provide the responses to each comment and question.

Also, we have  uploaded the revised paper that incorporates the reviewer comments and questions. All changes in the updated paper are highlighted in blue.

---

### Meta-Review · Area_Chair_gECx · 2023-12-09

**Metareview:**

The paper introduces Loco3D, a large-scale virtual reality dataset of multi-person interactions in indoor environments, aimed at improving AI modeling of human locomotion and interaction for applications like home robots and virtual avatars, demonstrating enhanced real-world trajectory synthesis. All reviewers recommend the rejection of the paper. The reviewers all agree that this is indeed a dataset paper but there are not enough supported experiments. After carefully reading the paper and the rebuttal, the AC agrees with the reviewers on rejecting the paper.

**Justification For Why Not Higher Score:**

The reviewers all agree that this is indeed a dataset paper but there are not enough supported experiments. After carefully reading the paper and the rebuttal, the AC agrees with the reviewers on rejecting the paper.

**Justification For Why Not Lower Score:**

N/A

---

### Decision · Program_Chairs · 2024-01-16

Reject